# An Accurate Quantitative X-ray Photoelectron Spectroscopy Study of Pure and Homogeneous ZrN Thin Films Deposited Using BPDMS

Mirenghi Luciana and Rizzo Antonella *

ENEA-Italian National Agency for New Technologies, Energy and Sustainable Economic Development, DivisionTechnologies and Processes of Sustainable Materials (PROMAS), Laboratory MATAS (Functional MAterials and Technologies for Sustainable ApplicationS)-Brindisi Research Center, S.S. 7 Appia Km. 706, 72100 Brindisi, Italy

* Correspondence: antonella.rizzo@enea.it

**Abstract:** A quantitative X-ray Photoelectron Spectroscopy (XPS) study has been undertaken on different experimental data sets of ZrN thin films deposited using reactive Bipolar Pulsed Dual-Magnetron Sputtering (BPDMS) on silicon/stainless steel substates, to obtain dense, pure and homogeneous coatings, free from morphological defects. Zirconium nitride (ZrN) occupies a central role within the class of transition metal nitrides (TMN) for its excellent properties, such as high hardness, low resistivity and chemical/thermal stability when its stoichiometric ratio is 1:1. Many deposition techniques, reported in the literature, tried to obtain oxygen-free and defect-free structures, but they proved a hard task. In this paper it has been demonstrated, using quantitative XPS, that stoichiometric, pure and homogeneous ZrN films have been grown at certain deposition conditions, optimized also via optional accessories mounted on the deposition apparatus. Almost all the films considered for microanalytical characterization resulted as completely oxygen-free, pure (with a lowest-detection limit of 1%) and homogeneous. Apart from these features, a stoichiometric ratio (N/Zr) close to one was calculated for six samples of the ten investigated, with a precision of $\pm$ 0.01. In this frame XPS, widely known for being a highly surface-sensitive technique (average depth resolution of 20–30 Å), and powerful for characterizing the chemical composition of materials, has been extensively employed to extract information both in the surface regions and in depth. A cluster ion beam Ar+ $_{2500}$ facility on our main XPS chamber has not proved adequate for depth-profiling acquisitions. Therefore, Ar+ ion sputtering was performed instead. To the best of our best knowledge, the results achieved in the present paper possess a level of accuracy never reached before. Rigorous calibration procedures before and during experimental spectrum acquisitions and a careful and scrupulous data processing using software CasaXps v.2.3.24PR1 were carried out to ensure a low percentage error. Progress has also been made for shake-up satellite extraction and interpretation from Zr 3d high-resolution spectra with the help of the literature milestones reported in the text. The total absence of oxygen inside most of the films prevented the formation of zirconium oxide compounds during deposition, which are generally resonant with the binding energy of the shake-up satellite peaks and hide them. A little summary about the experimental shake-up satellite peaks revealed and extracted from the Zr 3d region, after Shirley background subtraction and data processing, will be presented in the last subparagraph of the "Results" section for the ZrN samples analyzed. Figures of Zr 3d deconvoluted spectra for in-depth area analysis have been reported. The quantitative satellite contribution to the Zr 3d total area would not be included in stoichiometric calculations.

**Keywords:** ZrN; coatings; quantitative XPS; BPDMS; spectroscopy

## 1. Introduction

Among the wide class of binary nitrides, the transition metal nitrides (TMN), such as the AlN, SiN, TiN, VN, CrN, ZrN, NbN, MoN, HfN, TaN and WN ZrN films [1], occupy a

key-role for their attractive proprieties which are very useful for the production of different industrial products, such as protective coatings in the aerospace field, biomedical devices in the health sector and cutting tools in machinery [2].

The state of the art realization of increasingly performing ZrN films shows strengths and weaknesses for each laboratory methodology [3], i.e., some being less performant than others in obtaining defect-free structures, but good stoichiometry ratio N:Zr, one to one, as in the case of synthesis routes by wet [4,5]. A pulsed laser deposition technique instead produces dense and defect-free coatings but with a scarce N/Zr ratio (0.77). PVD depositions (i.e., reactive sputtering deposition, and dual ion beam sputtering) ensures quality films [6], but with a contamination of oxygen also at a high temperature of the substrate supporting the deposited film. HIPIMS and DCMS have been compared [7] for the case of the binary TiN system and a better performance of HIPIMS respect to DCMS came out. The publications of the last five years about TMNs films presented properties primarily dependent on the deposition method employed, partly by the working parameters [8–10], PVD or CVD [11–13] and also with original deposition methods for ZrN.

The misconception, often reported in the literature with covertly negative sentences, is that quantitative XPS analyses cannot be possible, whereas qualitative and semiquantitative XPS considerations sound more reliable because they are unaffected by data manipulation, which raises some disagreement. Quantitative XPS is in fact a very powerful aspect of data analyses, which needs good knowledge of the spectrometer components and their performances, long experience of data processing and finally good planning of the experimental campaign to produce data storage of high quality. The rather recently acquired BPDMS apparatus gave the authors the confirmation that among the deposition methods employed in the past, the new one can realize oxygen-free stoichiometric ZrN films. For further details about DPMDS it is suggested to refer to the authors' previous works. It was the groundwork for what has now become a comprehensive quantitative procedure study of ZrN films.

In the present paper, we are going to introduce quantitative chemical results, using XPS, obtained from three groups of ZrN films through accurate experimental and analytical procedures. Following the suggestions of Bears et al. [14], an area correction, after the Shirley background subtraction, has been carried out with three typologies of sensitivity factors: (1) Wagner tabulated sensitivity factors (SF), (2) theorical SF tabulated for Al mono source, and finally (3) experimental tabulated SF reported in PHI handbook of VersaProbeII 5000 Spectrometer (omitted results). The use of e-RSF of PHI Handbook were considered the most appropriate values to introduce in the narrow quantitative window of CasaXPS. Ar+ mild sputtering was fruitful in obtaining the chemical composition along the thickness of the samples and it was experimentally verified that it was preferable to the Ar+ 2500 cluster gun. The average stoichiometry of six of the ten specimens under investigation has resulted in the N/Zr being equal to 0.9.

## 2. Materials and Methods

### 2.1. Film Deposition

High-power reactive bipolar pulsed dual magnetron sputtering (BPDMS) system KENOSISTEC KSA 75 V, equipped with two pure (99.99%) Zr rectangular commercial targets in a mixture atmosphere, four nine of purity, made of Ar-$N_2$. The partial pressure ratio $N_2/ (Ar + N_2)$ was fixed at 0.1 while the pressure in the deposition chamber was 1 Pa. Two magnetron sources placed in a side-by-side geometry were plugged into a generator at a frequency of 80 kHz. The DC-pulsed power supply (TruPlasma Bipolar 4005) consists of the power source and a MOSFET switch section and has the role of reversing the two output powers: output 1 connected to the magnetron cathode 1 and output 2 to the second magnetron. A whole cycle of deposition corresponds to the period of sputtering and discharging for both the two targets. Target-substrate distance was fixed at 8 cm, and the substrate temperature was fixed at 300 °C. The mass spectrometer optional facility allowed us to display and acquire in real time the plasma emission spectra of

the atmosphere in proximity of the substrate in the range 200–950 nm during the glow discharge. It represented an additional value for the process control parameters such as the emission intensity ratios of nitrogen/zirconium, together with a cryogenic pump and a warming control of the targets. Three different groups of ZrN coatings were realized: (1) constant power (1.5 kW), (2) constant duty cycle (50%), (3) constant power (1.0 kW), as in Table 1.

**Table 1.** All the main deposition parameters are resumed for the ten samples deposited using BPDMS.

| SAMPLE Label | Duty Cycle | Power (kW) | T(Pulse) µs | Thickness µm | Frequency (Hz) |
|---|---|---|---|---|---|
| ZrN A | 50-50 | 1.5 | 12.5 | 2.94 | 80 |
| ZrN B | 33-33 | 1.5 | 8.25 | 2.50 | 80 |
| ZrN C | 25-25 | 1.5 | 6.25 | 2.70 | 80 |
| ZrN D | 20-20 | 1.5 | 5.0 | 1.97 | 80 |
| ZrN E | 50-50 | 2.0 | 12.5 | 2.5 | 80 |
| ZrN F | 50-50 | 1.5 | 12.5 | 2.40 | 80 |
| ZrN G | 50-50 | 1.0 | 12.5 | 2.20 | 80 |
| ZrN H | 50-50 | 0.5 | 12.5 | 2.50 | 80 |
| ZrN I | 50-50 | 1.0 | 12.5 | 2.0 | 80 |
| ZrN L | 33-33 | 1.0 | 8.25 | 2.0 | 80 |
| ZrN M | 25-25 | 1.0 | 6.25 | 2.0 | 80 |
| ZrN N | 20-20 | 1.0 | 5.0 | 2.0 | 80 |

*2.2. XPS Characterization*

X-ray photoelectron spectroscopy (XPS) technique is widely used for evaluating the chemical state of the near surface elements revealed on a material, through the determination of the chemical shift occurring for peak positions in relation to the species surrounding the atoms under investigation. Generally, the analysis performed is only qualitative and in better cases semiquantitative, but in this way it does not express its powerful potentiality. The possibility of extracting quantitative information from surface materials, with an accuracy of $\pm 1\%$, gives XPS analysis great potential. Of course, the intention to perform quantitative analyses should be clear to the analysts before operating and setting the experimental acquisition parameters. Calibration procedures and deep knowledge of the apparatus ensure a high degree of reliability, repeatability, and reproducibility of data at any time. In the present work a quite recent PHI VersaProbeII 5000 multi-equipped spectrometer was employed for XPS spectrum acquisitions. A monochromatic AlK$\alpha$ X-ray source with energy 1486.6 eV, power 12.5 W and 50 µm lateral resolution was employed for the acquisitions of all data sets. The polychromatic photoelectrons coming from the samples' surface were collected using a hemispherical analyzer set in FAT mode, whose operating parameters were chosen to optimize the acquisition of high-resolution spectra. In some particular cases, depth profiling along the whole thickness of the specimens under investigation could prove a fruitful method for overcoming the limits of the resolution depth, typical of XPS. Therefore, the performances of a sputter monoatomic Ar+ ion gun and cluster Ar+$_{2500}$ ion gun with maximum tension voltage of 20 kV were compared before acquiring depth profiling. The monoatomic Ar+ ion gun was chosen for sputtering all samples, taking a film of $Ta_2O_3$ as reference sample for calibrating sputtering time and thickness of eroded material according to working conditions. In the literature it is underlined that gas cluster sputtering guns are widely accepted for depth profiling of polymers samples, avoiding chemical damage or crosslinking but surprisingly in inorganic samples, and in particular on the commercial standard $Ta_2O_5$ (BCR-261T), preliminary tests showed that (i) the lower penetration depth does not lead to improved depth resolution when $Ta_2O_5$ is profiled, and (ii) even though cluster guns present a less preferential sputtering of the lighter elements in the first sputtering cycles, the use of 6 kV Ar+1000 ions onto $Ta_2O_5$ showed a gradual decrease in the O concentration [15]. This progressive loss of oxygen atoms as a function of depth was further enhanced when the experimental conditions

changed (higher tension and/or introducing sample rotation). However, an exhaustive comment for the choice of the mono ion gun rather than the cluster gun, for sputtering purposes, will be object of the section entitled "Discussions". The different ZrN films, deposited using the DPDMS method on conductive substrates, generally did not show any charging shift in spectra, despite sample ZrN D that displayed a 2.1 eV shift to lower binding energies suddenly correcting after spectra acquisition with respect to the lines of a ZrN-standard present in our laboratory database. With respect to the past study [16], the authors changed the fit function from SGL (30) to an "asymmetric Lorentzian" LA (a,b,c) included in the built-in functions of the CasaXps v.2.3.24PR1 software. LA (a,b,c) represents a kind of superset of Voigt function, with three variables and the possibility to include tail asymmetries on both sides in each synthetic peak considered. Nominally the "a" variable regulates the shape factor and must be properly chosen in combination with "b" and "c". From an operative point of view "a" regulates the tail behavior on the right side of the synthetic peak (i.e., at high BE), while" b" varies in the interval [1, 200] and influences both the tail factor on the left side and the FWHMs values. If properly tuned until an optimized residual standard deviation value is obtained (generally in the range [1, 3]) for numerical residual standard deviation-RSD),"b" gives good output results for the peak fits improving the quality of elaborated data and giving a lower margin of errors. Finally, "c" is an integer between 0 and 499, relative to the width of the synthetic peak it belongs to, but also influences the tail factor "a" and "b" variables together. It assumes widely different values during data elaboration. An accurate quantification of XPS data also depends on the ad hoc choice of the range of background subtraction. It absolutely remains a matter of primary importance, otherwise generating unreproducible results. The Shirley method has been chosen and the energy slot for correspondent core regions was assigned and fixed for background subtraction. In Table 2 the start/end points are reported for Zr 3d and N 1s core regions for 3 representative samples of the three groups of films, at surface and after sputtering (level 2). It is worthwhile to point out that the "start" and "end" points of level 2 remain identical for all the levels of depth profiling under the surface for both N 1s and Zr 3d regions. In Table 2 some experimental values for Shirley subtraction method are reported.

**Table 2.** N 1s and Zr 3d for three representative specimens of each group of samples.

| Sample ID | N 1s Start Point (eV) | N 1s End Point (eV) | Zr 3d Start Point (eV) | Zr 3d End Point (eV) |
|---|---|---|---|---|
| ZrN B group 1 level 1 (surface) | 393.180 | 401.302 | 176.684 | 189.625 |
| ZrN B group 1 level 2 (after sputtering) | 393.217 | 401.217 | 176.064 | 189.631 |
| ZrN H group 2 level 1 (surface) | 393.400 | 401.400 | 178.22 | 187.909 |
| ZrN H group 2 level 2 ( after sputtering) | 393.500 | 401.257 | 177.541 | 188.316 |
| ZrN N group 3 level 1 (surface) | 393.580 | 401.545 | 176.021 | 188.964 |
| ZrN N group 3 level 2 ( after sputtering) | 393.596 | 401.545 | 176.309 | 188.964 |

Peak area corrections were operated via tabulated e-RSFs (i.e., empirical sensitivity factors reported in the PHI handbook of the apparatus), and the correction factors depend

also on the specific transmission function of the analyzer. It was activated by ticking the "intensity correction" box and filling the empty field with the value $-0.6$. The theorical constrictions, such as the peak ratio in Zr 3d doublet fixed at 0.67 and the FWHM upper limit fixed at 2 in the satellite doublet, were introduced directly on the synthetic peaks in the "Narrow quantification window" of the software. A standard sputter Ar+ ion gun (model EX05) was employed for depth profiles using the bending geometry and shallow-condition angles (60–70° referred to the perpendicular axis to the surface of the sample).

In Table 3 the sputtering parameters are reported for samples of group 1. Similar conditions were applied to samples of groups 2 and 3.

**Table 3.** Sputtering parameters of the Ar+ ion gun related to an average pressure of $1.8 \times 10^{-6}$ mbar in the main chamber during gentle erosion (group 1).

| ZrN Sample ID | Sputter Energy (V) | Floating Enabled (V) | Filament Current (µA) | Bent Voltage (V) | Emission Current (mA) | Geometry Angle of the Gun (to the Normal) | Ion-Beam-Rastered Area (mm²) | Cycles |
|---|---|---|---|---|---|---|---|---|
| A | 2 kV | No | 2 | No | 15 | 70° | 3 × 3 (1 area) | 12 |
| B | 2 kV | Yes (−100) | 2 | Yes (1.76) | 3 | 70° | 3 × 3 (2areas) | 24 |
| C | 2 kV | No | 2 | Yes (70) | 4 | 70° | 3 × 3 (1 area) | 6 |
| D | 2 kV | No | 2 | Yes (70) | 4 | 70° | 3 × 3 (2areas) | 10 |

## 3. Preliminary Preparation Procedures for XPS Experimental Spectra Acquisitions and Definition of Quantitative Parameters of the Research

Before any activity, a calibration procedure was operated on the VersaProbe II 5000 PHI spectrometer employing a silver reference film, and the spectrometer was also sputter-cleaned to remove the adventitious contamination after air exposure. The mono X-ray source parameters were fixed as suggested by the vendor, to calibrate peak positions, but also to maximize their intensity values and optimize to full width at half maximum (hereon FWHM). After a survey spectrum acquisition check, the Ag 3d doublet was acquired. The Ag $3d_{5/2}$ line position was centered at $368.266 \pm 0.02$ eV of binding energy and the FWHM was 0.8 eV; these values ensure the correct position and the best energy resolution for each peak to be analyzed. Changing the constant pass energy of the analyzer with a step of 5 eV from 50 eV (for wide scans) to 20 eV (for narrow scans), it was possible to prove that the Ag $3d_{5/2}$ line kept its position and changes of the FWHM were registered. The uncertainty in reading a peak position is very low ($|0.02|$) for high resolution spectra. Therefore, after calibration procedures, we can trust the peak features we are going to acquire for films under investigation. The repeatability of data sets was ensured by the acquisition of the survey scan both at the beginning and at the end of each frame, in the acquisition program. Care was taken to transport the as-deposited films into XPS Laboratory, just a few meters ahead, using a glass jar in order to keep the surface clean from atmosphere contaminations. The objective of the research was to define via qualitative and quantitative XPS: (a) the phases present in each film according to different deposition conditions (for example the power voltage); (b) calculating the stoichiometry of the phase/phases revealed in the films, starting from the undersurface region, and going on by means of in-situ sputtering (alternating cycles of XPS acquisitions with sputtering ones); (c) acquiring core levels in the high resolution mode of at least two of the main adventitious contaminants, oxygen (O 1s) and carbon (C 1s), and defining the degree of purity of the films; (d) considering and verifying the homogeneity of the samples running parallel acquisition sets on different sputtered areas of the same film; (e) quantify the satellite contribution for each film, along its thickness, when we are sure there is no possibility that the satellite doublet can interfere

with any unwanted resonant oxide compound; (f) using a new-fit function with respect to the one normally used (SLG (30), built in CasaXPS function fit), it was possible to establish the role of satellite quantification to apply in stoichiometry calculation of the films, according to point (b) in the list of objectives here considered. A presentation of the main results obtained for each sample is reported in the following subparagraphs. It was necessary to match Zr 3d with N 1s for data quantification, and even though the start and end points for each elemental spectrum were kept fixed, sometimes it was necessary to move them a little in order to minimize the residual standard deviation values. This must be specified once and for all as a generally valid technicality, as mentioned in the introduction; the quantification of narrow areas, reported in the corresponding output window of the software CasaXPS, was achieved by correcting the raw areas with PHI e-RSF tabulated factors. The values 2.216 for Zr 3d and 0.47 for N 1s were typed in the box RSF because there is not a complete library yet for e-RSF factors. The experimental evidence indicated that sample A and its twin F, samples D, G and their twin I, and samples H, L, M and N present only one compound (ZrN), while the remaining samples B and C showed two distinct and co-present chemical environments along its thickness in N 1s region: the N-3 state, centered at $396.48 \pm 0.02$ eV associated with the $Zr_3N_4$ compound, and the peak centered in the range 397.3–397.6 eV associated with the ZrN contribution, as recently reported in the literature at 397.31 eV [17]. All the other films do not present attractive peculiarities for technological applications, as sample E was totally metallic in nature. The main results for each sample have been documented and commented on briefly.

### 3.1. Results of Sample ZrN A and Its Twin F

This sample presents, unlike all the others, two important features: (i) the ZrN compound can easily be revealed already on the as-received sample (Figure 1a level one), and (ii) a little amount of oxygen is trapped in the whole thickness of the sample (Figure 2 O 1s for level two, but similar spectra are available for the subsequent levels). In Figure 1a, the nitride peak position for N 1s stands at $397.30 \pm 0.02$ eV with a FWHM of 1.34 eV. After fit, Zr 3d showed instead two doublet-components, one centered at (179.5, 182.02) eV and associated to ZrN, the other at (181.99, 184.45) eV, representing the satellite's structure related to the zirconium nitride ($3d_{5/2}$, $3d_{3/2}$) synthetic peaks (refer to Figure 1b). A perfect correspondence between the experimental $\Delta = 2.43 \pm 0.02$ eV and the literature value has been found for the ZrN compound, where $\Delta$ represents the distance between $3d_{3/2}$ and $3d_{5/2}$ lines. The stoichiometry of the film at level two was calculated considering the ratio between N 1s total area and Zr 3d total area, but without the satellite's contribution. It resulted as being equal to 0.83 for level two, under stoichiometric in nitrogen when the N 1s peak position falls at 397.3 eV and 0.87 for level four when N 1s is peaked at 397.4 eV. Moreover, the Zr 3d of level four has a right-shift (as visible in Figure 1b right).

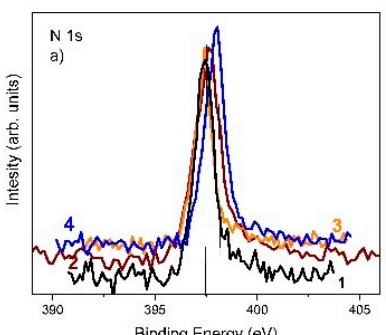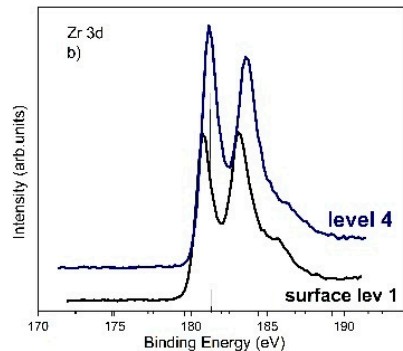

**Figure 1.** In (**a**) N 1s surface signal (1) is compared with the undersurface correspondent spectra (2,3,4) in sample ZrN A; in (**b**) Zr 3d surface signal has been compared with the in-depth spectra for the same element at level 4, in ZrN A.

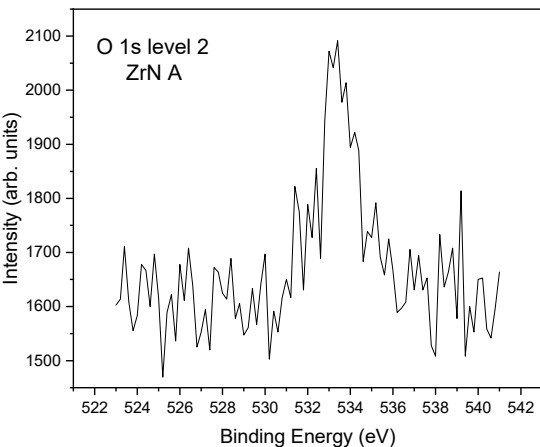

**Figure 2.** Oxygen contamination in sample ZrN A in-depth level 2.

A small amount of oxygen contamination [18], in this case, influences only the satellite contribution forming zirconium oxide. The peak fit with the LA (a,b,c) fit function has been pictured in Figure 3 (left and right) for level two of Zr 3d and N1s, respectively of sample ZrN A. Analogous situation was found analyzing the twin sample ZrN F.

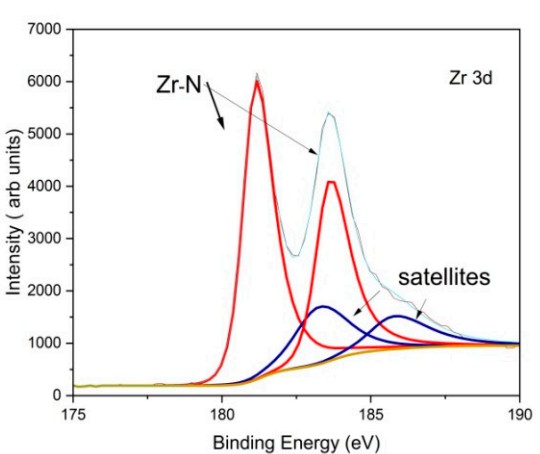
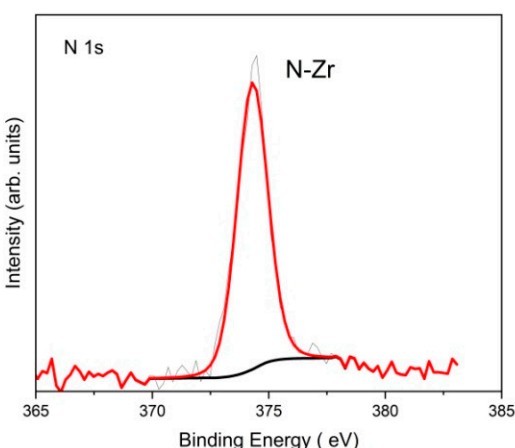

**Figure 3.** Peak fitting results for Zr3d and N 1s high resolution (HR) fitted spectra, for level 2 of sample A.

### 3.2. Results of Sample ZrN B

A different situation is found for coating ZrN B after peak-fitting separation. Here, the surface region shows a very thin top-layer (10–15 Å thick) of oxides (ZrOH/ZrO$_2$, eventually oxynitride) completely screening the nitrogen signal. The oxide top-layer disappears after a short sputter-clean of 2 min, and then acquisitions of the survey scans and the high-resolution spectra of N1 s, Zr 3d, O 1s and C 1s have been performed. The N 1s region shows two singlets, one at low BE (396.46 eV) associated to a 20.81% of Zr$_3$N$_4$ on the N 1s total signal, and the other centered at 397.35 eV associated with the ZrN compound. On the other hand, the Zr 3d region does not show the separation between the two phases (Figure 4 left side) but only the separation of satellite contribution from zirconium nitride ones. The statement [19] that a Zr 3d high-resolution spectrum cannot be separated into ZrN and Zr$_3$N$_4$ contributions by fit is widely shared. In fact, when eventually both phases are present, they overlap. The right side of Figure 4 reports the results after a peak fit of N 1s where N-Zr bonds prevail but a low BE peak is present and attributed to Zr$_3$N$_4$ [20]. At this point it becomes very interesting to study in detail sample ZrN B.

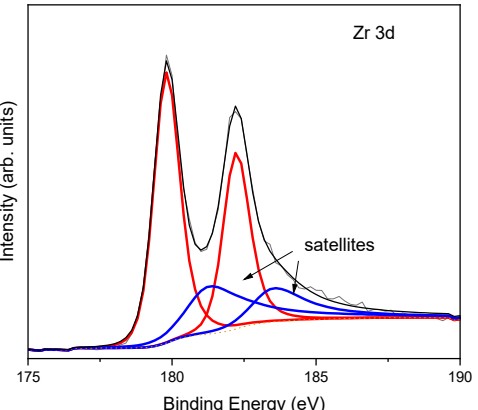
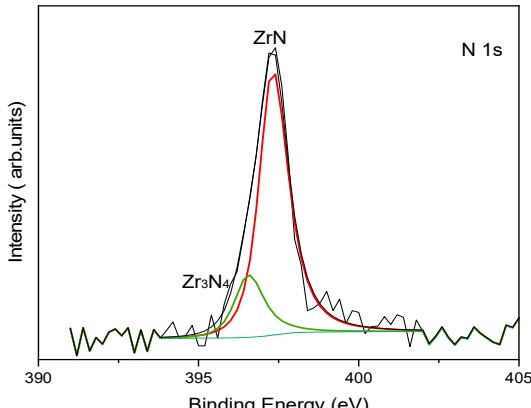

**Figure 4.** Peak fitting results of Zr 3d and N 1s HR (high resolution) regions at level 2 from the surface for sample ZrN B.

A complete sputtering depth profiling using theAr+ ion gun composed of 24 points from the surface was constructed at a constant temporal step. The sputtering conditions have been reported in Table 3. The in-depth acquisitions were repeated on two different areas of 9 mm$^2$. In Figure 5 the first 13 points are reported, cutting all the second part of the depth profiling for the sake of simplicity. Looking at Figure 5, the $Zr_3N_4$ percentage (N1s II) stands always below the ZrN percentage that is 10% (N 1s I) and the Zr 3d nitride doublet includes both the ZrN and $Zr_3N_4$ indistinguishable contributes. N 1sI indicates the stoichiometric nitride and N 1sII the over-stochiometric nitride separated on the N1s HR spectrum. The trend of satellite percentages (with respect to the overall Zr 3d HR spectrum) is indicated: its average value stands at 20–25% with respect to the remaining signal associated with nitride species.

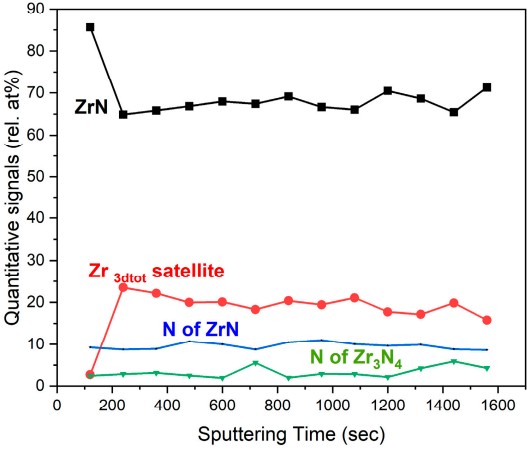

**Figure 5.** Sputter depth profiling of sample ZrN B for the four subcomponents extracted after peak fit and indicated on each graph.

### 3.3. Results of Sample ZrN C

The coating, labelled as ZrN C, presents a very noisy wide scan signal, related to the undersurface region, as it can be seen looking at the survey scan reported in Figure 6 (top spectrum). However, it is clear that no oxygen contamination is detected either of the other undesired chemical species. Ar+ implantation for sputtering is not yet revealed, since it is also difficult to discern with such a high noise intensity. The noisy trend of the spectrum can be ascribed to the microstructure of the sample which presents during scanning electron-microscopy observations performed by the authors and published elsewhere [16], observations that showed a microstructure with grains of large dimensions

(around 50÷150 nm) together with small grains of a diameter around 13 nm. In Figure 6 (bottom left), the region of Zr 3d after fit is pictured at the same depth from the surface of the survey scan of Figure 6 (top spectrum). N 1s high resolution peak has also been reported in Figure 6 (bottom right) Here the intensity (cps) of nitrogen respect to zirconium seems very low, without taking into account the different cross section of the two elements and the other correction factors as the escape depth and the transmission function of the analyzer. Calculation of the stoichiometric ratio [21,22], like all the samples investigated up to now, gave for sample ZrN C a N/Zr value equal to 1.63, confirming the presence of $Zr_3N_4$ besides the predominant ZrN component.

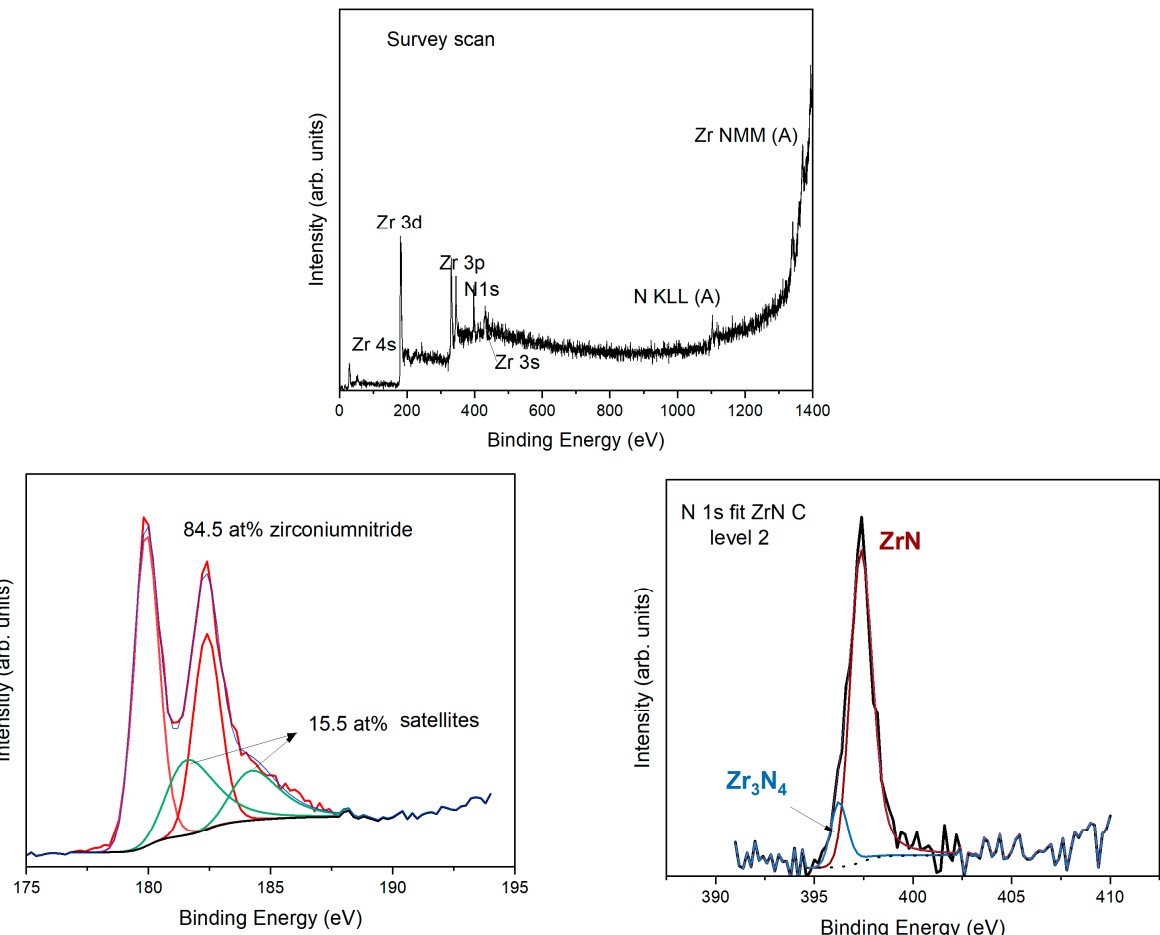

**Figure 6.** Survey scan of sample ZrN C (top spectrum) after Ar+ etch-cleaning of 2 s (level 2 of acquisition) with identification of both photoelectrons and Auger peaks (A). (Bottom left) Zr 3d region after fit and (bottom right) N 1s region after fit for sample ZrN C.

### 3.4. Results of Sample ZrN D

Although the situation seems similar to the coatings previously illustrated, as sample ZrN D shows in the first undersurface layer, only the formation of theZrN compound (see Figure 7a,b) shows it is not the case. In fact, an electrostatic charging at level two of +2.1 eV was shown by the N 1s peak (meaning it is not a good conductor of electrons), and also an over-stoichiometry in nitrogen was assessed calculating the N/Zr which resulted equal to 1.14 compatible with the 1.16 expected when there is an average content of nitrogen shared between ZrN (1:1) and $Zr_3N_4$ (1:1.33). Briefly, from level 2 onwards, the N 1s peak becomes asymmetrical on the low-energy side because of the presence of $Zr_3N_4$ in co-presence with ZrN. All along the depth profiling the co-presence of the two compounds can be confirmed. In Figure 7a,b Zr 3d and N 1s spectra after fit with the LA (a,b,c) function are reported.

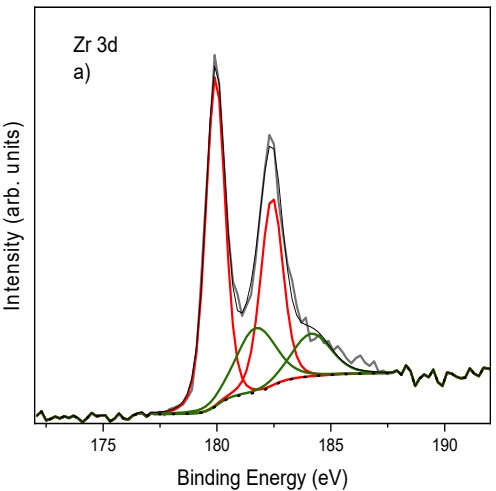
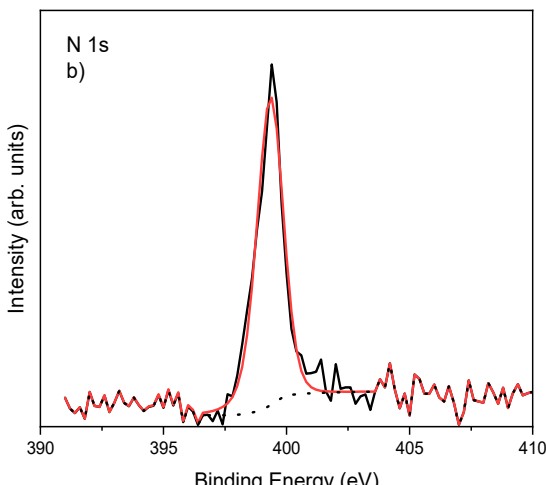

**Figure 7.** Zr 3d region (**a**) and N1s region (**b**) of sample ZrN D after peak fitting of level 2.

### 3.5. Results of Sample ZrN E

This sample resulted in being totally metallic in nature; in fact, the Zr $3d_{5/2}$-line falls at $178.1 \pm 0.02$, corresponding to the Zr-Zr bonds (online sr.data.gov/xps/ accessed on 24 October 2022), while the spin-orbit splitting is equal to 2.4 eV. No nitrogen signal was revealed. During deposition at 2000 W arc discharges were observed. The deposition parameters reveal, in this case, as being inadequate for growing a ZrN coating.

### 3.6. Results of Sample ZrN G and Its Twin I

Sample G belongs to two groups (two and three, see Table 1, where the twin is also indicated) of coatings using BPDMS: one having a constant duty cycle and the other having a constant high-power of 1000 W. No chemical changes were detected along the whole depth profiling. The first level after Ar+ etching-cleaning is reported in Figure 8 and depicts the Zr 3d and N 1s regions after fit; no additional features or artefacts coming from the Ar+ sputtering were observed. The extra features and artefacts introduced after the sputtering cycles reported in the literature here have not been observed; either charge correction or Y-scale rescaling was operated on rough data. Sample ZrN G is a homogeneous coating from a chemical point of view. The peak-fitted N 1s region mainly presents a unique component centered at 397.4 eV, typical of N-Zr bonds. Concordantly, the peak fitted Zr 3d region, corresponding to the same level of N 1s, evidences that only the ZrN compound and related satellite have been detected (Figure 8).

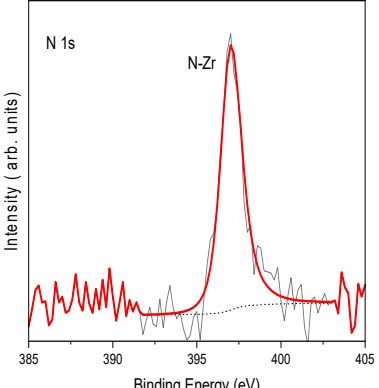
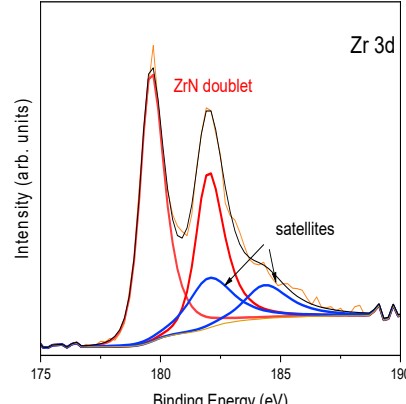

**Figure 8.** Peak-fitting results of Zr 3d and N1s HR regions level 2 sample ZrN F group2.

### 3.7. Results of Samples ZrN H, ZrN L, ZrN M and ZrN N

According to the intended objective of this work, the subset of samples ispresented as follows; witness the example of the desired results. In fact, many answers are found to the challenges listed in Section 3 (from point a to f).

In Figure 9a, a peak comparison for the three main elemental regions of samples ZrN H, L, N is reported, referring to raw data without any charge correction or manipulation. Sample ZrN M shows many similarities with sample ZrN N and, therefore, it was cut out. Care was taken with the BE range of the X-axis: it was unchanged for each correspondent chemical species to make a reasonable comparison. They were chosen in the following intervals expressed in Binding Energy (BE): [172.0, 192.0] eV for Zr 3d, [390.0, 408.0] for N 1s, and [520.0, 540.0] eV for O 1s. Preliminary and general considerations can be deduced by looking at Figure 9a where the surface behavior of samples H, L and N is reported. Oxygen is revealed on the surface region of all samples, forming native oxides with zirconium, which is $ZrO_2$ -like and centered at 530.2 eV on the O 1s spectrum, and at about 180 eV on Zr $3d_{5/2}$ line, while the hump of O 1s centered at 531.4 eV and is associated with the ZrON compound.

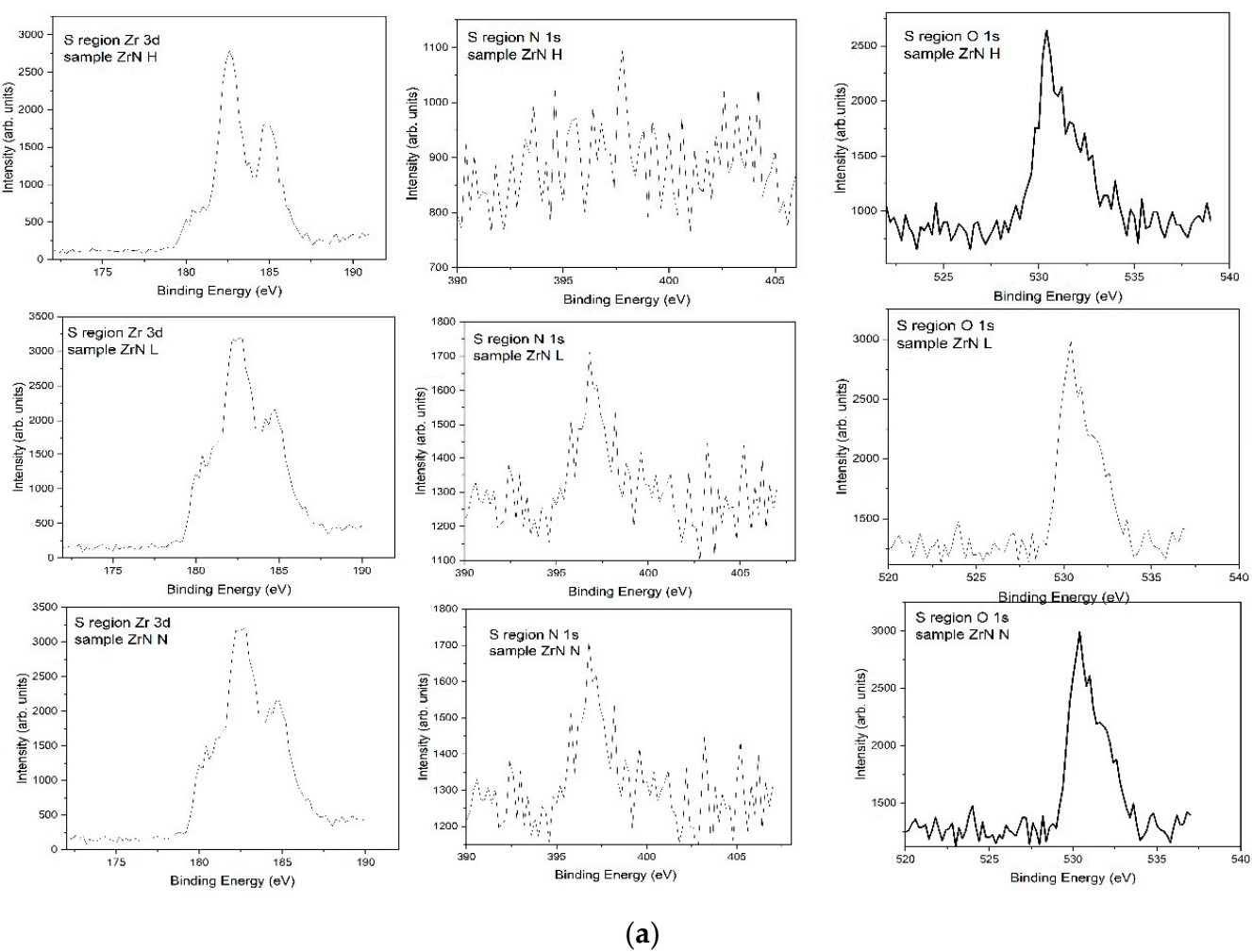

(**a**)

**Figure 9.** *Cont.*

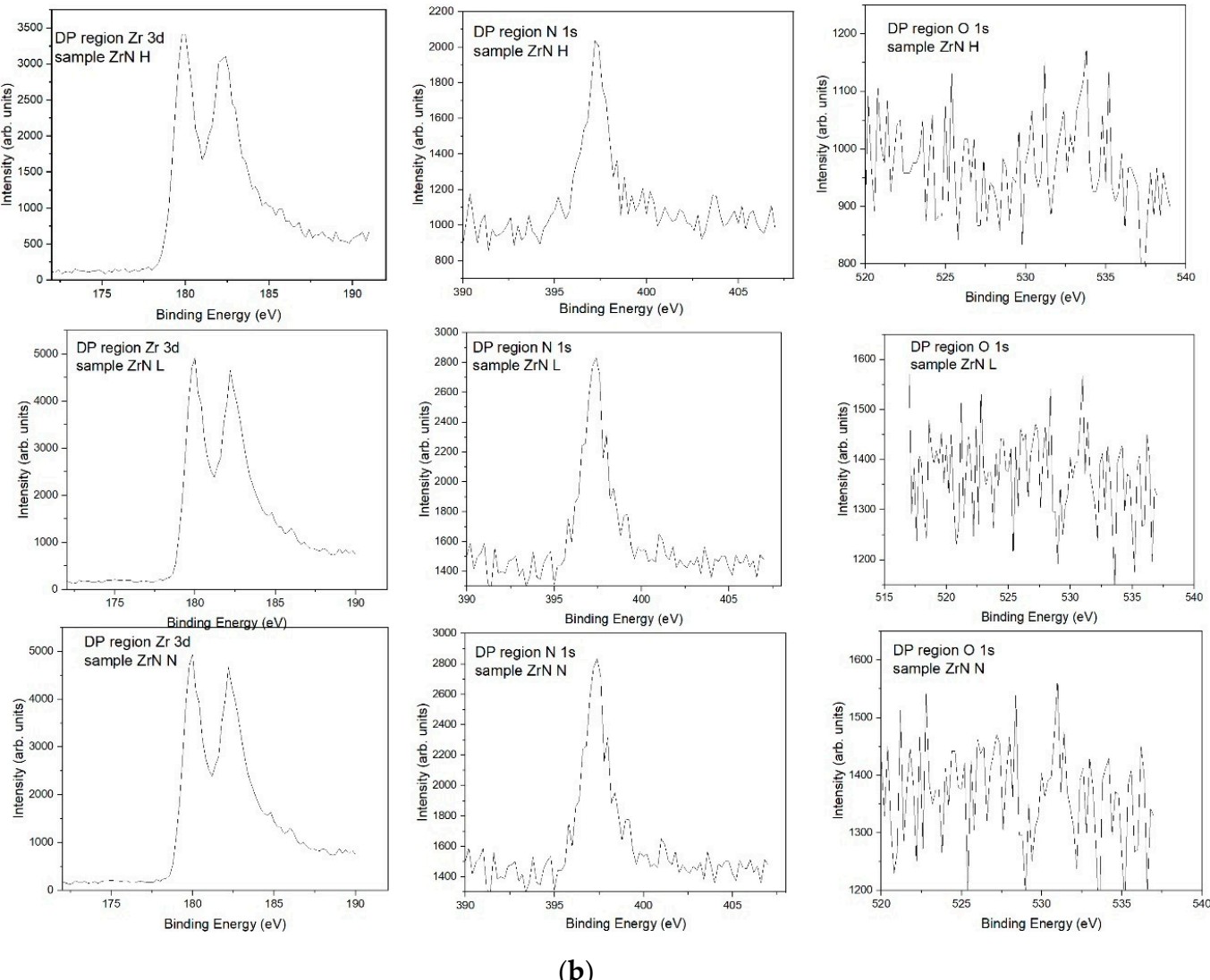

(**b**)

**Figure 9.** (**a**): Surface high-resolution spectra of Zr 3d, N 1s and O1s for samples ZrN H, L, N. (**b**): In-depth acquisitions of high-resolution spectra of Zr 3d, N 1s and O1s for samples ZrN H, L, N after two sputter-cycles (labelled as DP region).

The qualitative surface scenario found is common to many transition metal nitrides obtained with different deposition techniques: a thin-layer oxide/hydroxide (eventually oxynitride) is formed after air exposure and the nitrogen signal (N 1s region) cannot be visible (sample H) or distinctly revealed on the surface of samples L and N according to the thickness of the oxide/hydroxide layer. Concordantly, the Zr 3d peak shape changes from sample H to L and N samples, due to the only contributions of oxide/hydroxide in the first row. The situation after in-situ sputtering has been represented in Figure 9b. The oxygen signal goes down to zero for all samples and the Zr 3d peak shape and position can be associated with zirconium nitride. At the same time, the N 1s line stands centered at $397.4 \pm 0.02$ eV with a typical tail from the high-binding-energy side [23]. The absence of oxygen as well as any other kind of contamination for the in-depth samples helped us [24] proceed to a quantitative peak-fit deconvolution in order to extract the synthetic peaks, contributing to the overall experimental peak shapes and the reading of their positions.

Finally, XPS quantitative results, for samples L, M and N are reported in Table 4 for the first parts of their complete depth-profiling. The results obtained from each of the two fit functions SLG (30) and LA (a,b,c) are also compared.

**Table 4.** ZrN samples quantitative results.

| Sample ID/Level from Surface | N/Zr SLG (30) * Fit Funct. | Value of Residual STD for Zr 3d * | Value of Residual STD for N 1s * | N/Zr LA (a,b,c) * Fit Func. | Value of Residual STD for Zr 3d ** | Value of Residual STD for N 1s ** |
|---|---|---|---|---|---|---|
| ZrN L level 2 | 1.08 | 2.168 | 0.6242 | 0.81 | 1.890 | 1.023 |
| ZrN L level 3 | 1.09 | 2.894 | 0.7336 | 0.87 | 2.356 | 1.167 |
| ZrN L level 4 | 1.19 | 2.563 | 0.7098 | 0.87 | 1.794 | 1.065 |
| ZrN L level 5 | 1.18 | 2.739 | 0.7779 | 0.84 | 1.883 | 0.917 |
| ZrN M level 2 | 1.10 | 2.75 | 1.008 | 1.16 | 1.557 | 1.205 |
| ZrN M level 3 | 1.13 | 2.868 | 1.176 | 1.10 | 1.488 | 1.187 |
| ZrN M level 4 | 1.32 | 2.636 | 1.21 | 1.09 | 1.656 | 1.22 |
| ZrN M level 5 | 1.05 | 2.456 | 1.142 | 1.05 | 1.345 | 1.373 |
| ZrN N level 2 | 1.23 | 2.276 | 1.186 | 0.91 | 0.955 | 1.009 |
| ZrN N level 3 | 1.27 | 2.617 | 1.089 | 1.07 | 0.937 | 1.089 |
| ZrN N level 4 | 1.22 | 2.29 | 1.136 | 0.89 | 1.266 | 1.113 |
| ZrN N level 5 | 1.29 | 2.157 | 1.152 | 0.83 | 1.416 | 0.8715 |

** refers to the SLG (30) fit function and is recalled in the respective coloumns "Value of residual STD for Zr 3d and N 1s", while * refers to LA (a,b,c) Fit Function for the correspondent two last coloumns labels.

Calculated stoichiometries (N/Zr ratio, labelled R) for sample ZrN H are reported for completeness for levels n = 2,3,4,5 from the surface (level one) for the fit built-in function LA (a, b, c) as in Table 4: for n = 2 R is 0.92; for n = 3 R is 0.99; for n = 4 R is 1.03; for n = 5 R is 1.19; for n = 6 R is 0.94. A further confirmation of the realization of ZrN stoichiometric coatings comes from the final XPS quantitative results reported in Table 4 for the first four layers of the complete depth-profiling.

After some qualitative considerations on N 1s high-resolution XPS spectra parameters (position, FWHM and tail behavior from the high BE side) it can be asserted that samples L, M and N after a gentle sputter-cleaning to remove surface contaminants (i.e., at level 2 of the acquisition) present only one compound, identified as ZrN, presumably with a c-ZrN NaCl-like structure as deduced using x-ray diffraction (XRD) spectra (not reported here) performed on the same samples analyzed using XPS. The analyses were carried out using an x-ray diffraction system (XRD, EMPYREAN, PAN analytical) equipped with a copper anode (40 kV\40 mA). XRD apparatus was set up in a particular geometrical configuration which assures an intense monochromatic beam for a good signal-to-noise ratio and allows us to operate in a parallel-beam geometry suitable for asymmetrical and symmetrical reflection measurements. The phase identification was performed according to the well-known reference databases JCPDSICDD 2000. The experimental features compared with the database assessed the presence of a NaCl-like cubic structure for the ZrN films. A cascade graph displayed in Figure 10 (left and right) shows the comparison for N 1s and Zr 3d lines, respectively. The acquired data give evidence of the formation of a unique chemical environment for the three samples, presenting the main peaks in the same position.

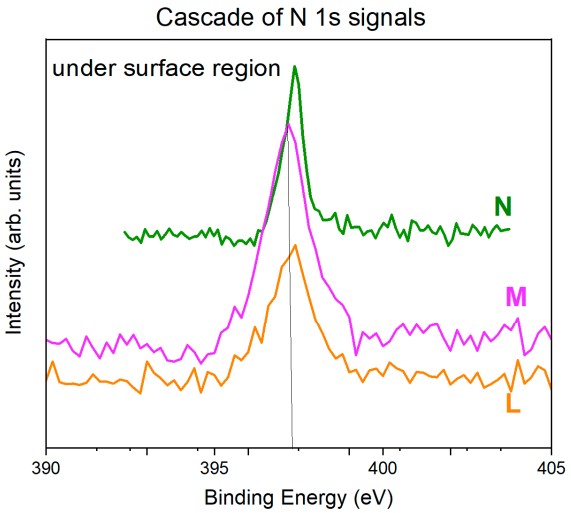
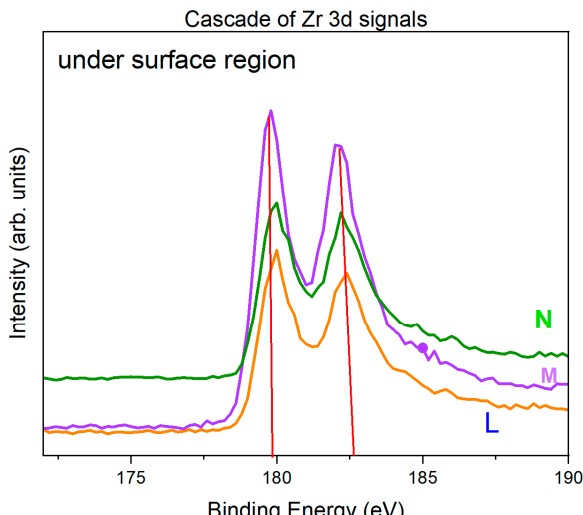

**Figure 10.** Comparison of N 1s (left) and Zr 3d (right) XPS regions, for the under-surface region (level 2), for the three different stoichiometric ZrN coatings: ZrN L, ZrN M and ZrN N.

The formation of ZrN was associated also with sample ZrN H, whose fits of Zr 3d and N1s are reported in Figure 11a,b, respectively. The main output parameters after fit (i.e., Zr 3d peak positions, main doublet distance Δ = 2.5 eV, N1s peak position centered at 397.40 eV) confirmed, also for sample ZrN H, the formation of the ZrN compound.

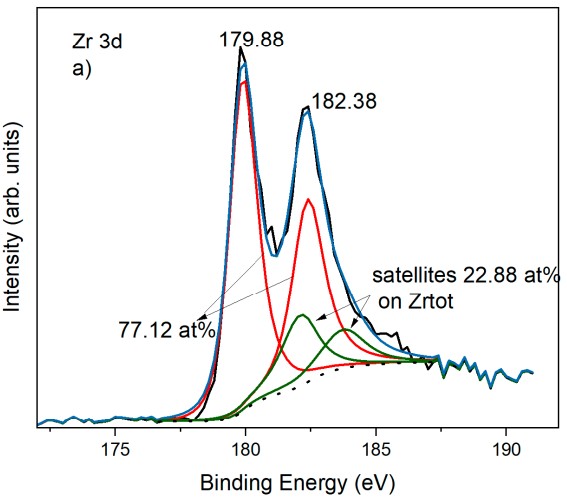
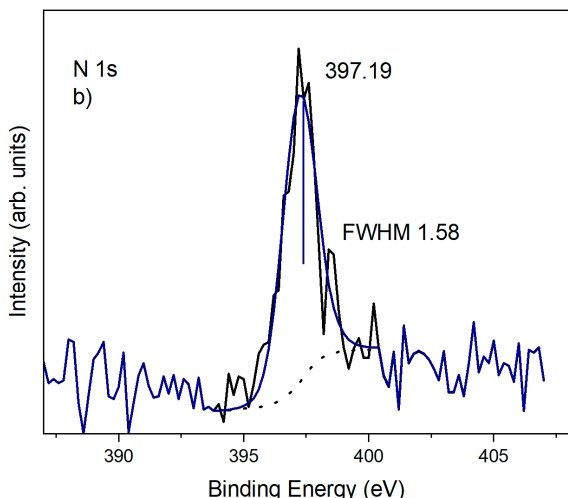

**Figure 11.** Zr 3d, (**a**) and N 1s (**b**) fitted regions at level 5 from the surface, after Ar+ in situ sputtering of sample H.

### 3.8. Results and Considerations about Satellite Peaks of High Resolution Zr 3d in All Samples Investigated

Although 53 years have passed since the first publication about TiC and VC satellite structure [25], many questions still remain [26].

As correctly stated in the literature [25], shake-up satellites in core-electron spectra of transition-metal complexes result from a charge-transfer transition after the photoelectron ejection, and if charge transfers are observed in the absorption spectra, they will appear also on XPS spectra, as long as allowed by the selection rules.

A further difficulty comes from the nature of the metal and the ligands and the symmetry of the structure under consideration. Moreover, when the overlapping of the satellite peak with a photoelectron line (or spectral feature) occurs, the solution of the

puzzle becomes very difficult to reconstruct. Often satellite positions and their intensities have reported a certain firmness, but in contrast to each other. The topic is controversial [26] and far from simple. As referred by Herrera-Gomez et al. [27], the need for an accurate peak-fitting on the transition-metal region is welcome to point out humps or hidden satellite features, considering both branches of the first row of transition metals and not only the main branch, as reported countless times.

The authors believe that, apart from the specific characteristics of the system under consideration, an accurate peak-fitting procedure on both branches of the transition metal doublet does not close the cycle. A theoretical model is needed for the specific complex system under investigation, Pauly N et al. [28] established by quantitatively interpreting the shape and the intensities of shake-up structures originating from the photoexcitation process with their own model on silver and gold. In our case, the system consists of metallic stoichiometric c-ZrN and no tailored theorical model to apply was found. Referring to an experimental paper [17] where the authors refer that their report "is the first on satellites in the Zr 3d spectrum for ZrN films", the authors have been encouraged to go on with Zr 3d satellite deconvolution and interpretation without any theorical support.

The starting points for the peak-fitting processing of each Zr 3d spectrum of the depth profiling for ZrN films analyzed (only six of those ZrN films were single binary compounds) are:

(1) the separation of shake-up satellites from the line-Zr 3d-doublet associated with zirconium nitride was fixed at + 2.5 eV [16], where the sign indicates the shift at higher binding energies with respect to the main peak (j = 5/2 as well as j = 3/2). The reference value was also supported via an internal laboratory database, the TiN system, about which the authors know well, and presents affinities with the ZrN system [21];

(2) no constrains were imposed to the satellite's components differently from the main doublet to better observe where the peak position falls for both shake-up satellites;

(3) for area quantifications after peak fit, the same RSF factor of the main doublet was used and the LA (a,b,c) was employed as built-in function of CasaXPS, but the a,b,c parameters were left free in the reiterations (4–6) applied to the deconvolution process.

It was observed that their quantification features changed according to the sample structural order and the chemical composition of the pure ZrN or pure ZrNx with x < 1 and x > 1. The peak position of the shake-up associated with the photopeak Zr 3d stands at 182.4 ± 0.02 for stoichiometric films M, N, but changes for sub stoichiometric samples (i.e., L) or where there is little oxygen in-depth detected. The line movement is represented in Figure 1b and a constant behavior for peaks position of the two satellites was revealed along the depth profiling for each film, suggesting a structural and chemical stability for each film.

The total area percentage goes from 20–24% of the total Zr 3d signal in sample ZrN A (group 1) to 21–31% for ZrN G (group 2) and finally from 30–37.5% for sample ZrN N (group 3) as reported in the histogram of Figure 12. ZrN stoichiometric, pure and homogeneous films present a higher quantitative contribute of satellites respect to sub or over stochiometric samples. In the quantitative N/Zr ratios, the satellite's contribution was not included, as should happen more generally for all transition metals' composites. Further details about shake-ups satellites in the Zr 3d spectra of theZrN system will be considered in another publication.

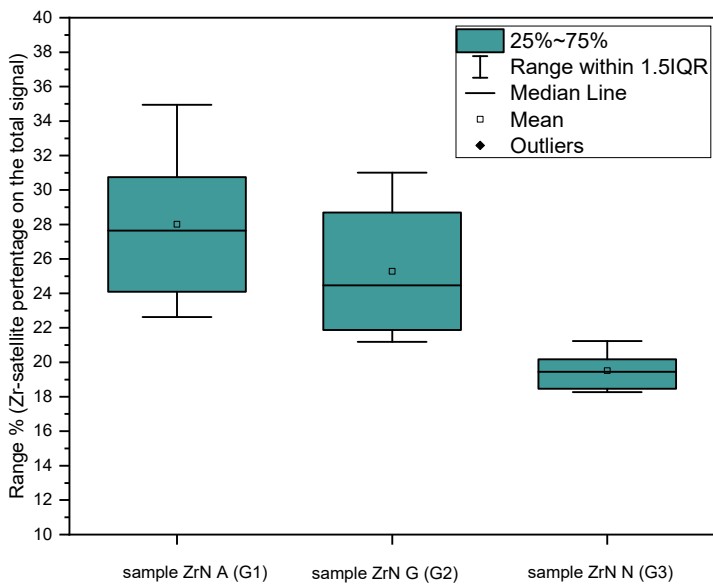

**Figure 12.** Histogram of Zr 3d satellite comparison among three representative samples A, G and N.

## 4. Discussions

Great efforts have been made to study the chemical features and the composition of ten ZrN coatings deposited using BPDMS, through quantitative XPS, using a new built-in fit function, denominated LA (a,b,c), which succeeded in taking into consideration asymmetries and tail effects during the fit of raw data. Following the practical guidelines for quantitative XPS recently published by Alexander C. Shard [22], the acquisition and storage of confident, raw XPS data allowed the authors to obtain accurate quantitative results about stoichiometry, degree of purity and homogeneity of ZrN films, using the software tool CasaXPS v.2.3.24PR1.

It was also possible to compare the main quantitative results obtained through the fit function SLG (30) with those of LA (a,b,c) with optimized parameters (cfr Table 4) by using the Shirley background subtraction method applied, fixing the same start-end points in both cases for a rigorous comparison. It was proved that the SLG (30) built-in function systematically over-estimated the N/Zr ratio, because there is a great margin of error in the Zr 3d area calculation with the SLG (30) rather than with LA (a,b,c) function, due to the complicated fine structure and peak shape of Zr 3d regions. It is easy to notice that the residual STD values of Zr 3d are always higher than 2.2 when the fit function corresponds to SLG (30). On the other hand the residual STD for LA (a,b,c) has an average value of 1.499, proving, in part, a better quality of the deconvolutions obtained by selecting the LA (a,b,c) peak function.

The main difficulty we came across was to reconcile the elaboration of a huge amount of data acquired with the need to reach the highest level of accuracy at the expense of long processing times. We performed surface and in-depth acquisitions, the latter obtained through Ar+ ion sputtering, according to working parameters reported in Table 3. Some interesting samples were undergone for depth-profiling in different areas in order to verify theirs homogeneity, as for sample N. Another important result was achieved during a preliminary comparison between the $Ar+_{2500}$ cluster gun performance and the monoatomic argon (Ar+)-sputtering ion gun on sample ZrN A, the accessories being both present in the main chamber of the VersaprobeII 5000 spectrometer. Several proofs were attempted both with a 6 kV and an 8 kV for the cluster gun and all the other parameters suggested by the vendor. On the other hand, the Ar+ cluster gun was settled at 3 kV, 2 μA of filament current and 15 mA of sample current. The cluster gun after several cycles of acquisition/sputtering showed a similar situation to the as-received sample, while the monoatomic gun gave appreciable results [15]. The reading key is considering the sputtering not only as a ballistic

process but also as a more complex combination of mechanisms for all the high temperature spikes at the surface carrying to the infusion of elements on the solid surface. On these solid foundations of cluster ions, sputtering theory and the experimental preliminary comparison of the two guns, the VersaProbe II 5000 and monoatomic Ar+ ion gun, was employed to perform depth profiling. The argon ion sputtering theory [29] allowed us to conclude that after just a few layers of sputtering where the lighter element (N) is preferentially removed, a steady state is reached. Therefore, the under stoichiometry revealed is not an artefact due to the nitrogen-preferential sputtering effect, but the actual elemental deficiency. There is no doubt that the group-three samples, grown at 1 kW, represent examples of ordered composition, being cubic in structure and NaCl-like, with stoichiometries near the ratio 1:1 and a measured hardness in the range 30–35 GPa [16].

A comparison between ZrN M and ZrN C samples, as well as between N and D, or M and C, pointed out that they are completely different. This was expected because the deposition processes have cross features (i.e. are similar to the same single phase of ZrN, but not the same quantitative features, as deduced by reported results).

Shake-up satellite peaks were quantified to correctly consider the Zr 3d area contribution in stoichiometry calculations. Their features (i.e., position and percentage intensity) varied according to the sample structural order and its chemical composition (ZrN or ZrNx with x > 1 or x < 1). The peak position of the shake-up associated with the Zr $3d_{5/2}$ stands at $182.4 \pm 0.02$ for stoichiometric films M, N but changes for sub stoichiometric samples (i.e., H 181.79 eV) or where there is little oxygen; the latter moves the satellite line as pictured in Figure 1b; a constant behavior for the peak position of the two satellites was revealed along the depth profiling for each film, suggesting a structural and chemical stability.

## 5. Conclusions

Ten ZrN coatings deposited using reactive bipolar pulsed dual magnetron sputtering, at different conditions, were quantitatively characterized by x-ray photoelectron spectroscopy both on the surface region and along their thicknesses, with a high level of accuracy for data elaboration. To the best of the knowledge authors' here, for the first time a comparison of the results for SLG (30) and LA (a,b,c) peak-fitting functions is presented for a TMN binary system. The SLG (30) built-in function systematically over-estimates the N/Zr ratio, because in calculating the synthetic Zr 3d areas SLG (30) it shows a greater discrepancy with experimental curves rather than the LA (a,b,c) function. On N 1s the two functions gave similar results for both built-in functions. The area discrepancies for the Zr 3d spectrum can be related to its complicated peak shape (cfr. Table 4).

The coatings analyzed were all free from oxygen and any other contamination, except for sample A, presenting a very slight oxygen signal along its thickness. Six of the ten samples analyzed are nearly stoichiometric (1:1 ratio) with an average excursion in the range [0.7, 1] along the depths. This desired result has taken years; also, for the optional facilities present on the deposition apparatus, such as for example the mass spectrometer, this allowed us to acquire the plasma-emission spectra in the proximity of the growing film, thus allowing us to control and avoid oxygen contamination. Twin-deposited samples possess the same chemical features as their brother ones, demonstrating the repeatability of the films under the same experimental conditions.

A monoatomic Ar+ cluster gun was preferred to the cluster $Ar^+_{2500}$ ion gun for depth-profiling acquisitions. The choice was motivated because the cluster ion gun source showed a low yield of nitrogen sputtering inside the sample, probably due to the thermal effects besides mere ballistic phenomena, and also the lower penetration depth respect to the monoatomic source.

Shake-up satellite peaks of transition metal revealed in XPS core-level spectra have generated confusion and controversies in the literature but their positions and intensities give important information about the system under investigation. In this paper, the authors were faced with shake-up satellites on Zr 3d high-resolution spectra to accurately calculate

the stoichiometries of oxygen-free ZrN films, neglecting the satellite contribution from the Zr 3d total area.

It was observed that shake-up satellite quantitative features changed according to the sample structural order and its chemical composition. In ZrN films (if ZrN or $Zr_3N_4$), a shift was observed in the photopeak position associated to the main line Zr $3d_{5/2}$. It stands at 182.4 $\pm$ 0.02 for stoichiometric films M, N but changes for over-stoichiometric samples (i.e., B) at 181.79 $\pm$ 0.02 or where there is little oxygen in-depth, as in sample A (cfr. Figure 1b). A constant peak position of the two satellites was revealed instead along the depth profiling of each film, suggesting homogeneity in the areas analyzed.

Some of the coatings realized with a new deposition technique and characterized microanalytically using quantitative XPS were shown to be pure, repeatable and with N/Zr ratios of 1:1, suggesting their importance and application in many technological fields [30].

**Author Contributions:** Conceptualization M.L. and R.A. All the rest M.L. All authors have read and agreed to the published version of the manuscript.

**Funding:** National Project entitled "Close to Earth" acronym CLOSE- Project code ARS01_00141.

**Institutional Review Board Statement:** Not applicable.

**Informed Consent Statement:** Not applicable.

**Data Availability Statement:** Simply contacting luciana.mirenghi@enea.it.

**Acknowledgments:** The authors wish to warmly thank Neal Farley for their professional, usefuland always kind support in helping us tounderstand the multifaceted world of the software CasaXPS, even though a long time has passed since the first show of interest in the software package.

**Conflicts of Interest:** The authors declare no conflict of interest.

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
