# Peer review of "An Accurate Quantitative X-ray Photoelectron Spectroscopy Study of Pure and Homogeneous ZrN Thin Films Deposited Using BPDMS"

_applsci, doi:10.3390/app13031271_

Round 1

Reviewer 1 Report

Authors have reported the Quantitative X-ray photoelectron spectroscopy of pure and homogeneous ZrN thin films deposited by High Power reactive Bipolar Pulsed Dual Magnetron Sputtering (HP BPDMS): an accurate study. This manuscript needs minor revision before publication.

1.      Here author discussed the magneto sputtering techniques for thin film deposition; author can compared their data with the reported data in which different technique for thin film deposition is used.

2.      As author discussed thin film deposition using magneto sputtering in the manuscript but purity of the sample and structural properties of the sample must be needed. So it is mandatory to characterize the sample using EDAX and XRD.

3.      The language of the paper must be needed to improve. Grammatical mistakes must be also nullified.

4.      The notation for each figure throughout the manuscript must be same. Also the format of all the figures in manuscript must be same. i.e. author used “figure” in picture format but in discussion part they use “fig. (Left or Right) and it is inappropriate. so author needs to be mentioned appropriate figure name and figure no.

5.      As author discussed in whole manuscript about peak fitting but they do not mention anywhere which fitting they used.

6.      Overall the scientific content of the paper is good. Therefore this manuscript can be considered for publication if the author can resolve these comments.         

Author Response

Anwers to Referee 1 (identified by peer-review-24403676 document signed)

Kind Referee 1,

we are going to ansewer to your  comments and suggestions to the authors about manuscript ID-appsci-2041668

English language style, grammatical and spelling mistakes  of the original paper have been corrected. Changing the English style in many sentecnces has also improved the quality of the whole paper and vanished some misunderstandings.

The reserch design has been improved introducing paragraph 3in the text with the Title “Preliminary preparation procedures for XPS experimental spectra acquisition and definition  of quantitative parameters for the research”

  • The authors have accepted to compare their own data of ZrN films with data published on ZrN realized with different deposition methods. This improvement carried out the additionof one sentence in the “introduction” paragraph.

  • The mandatory suggestion to characterize the samples of ZrN by EDAX (for purity information) and XRD (for structural properties) is very reasonable, in order to confirm the XPS results. Some results and considerations will be added in the “discussion” paragraph of the manuscript because we performed XRD (which gave a c_ZrN structuref or exclusive ZrN compound with stoivhiometry near to 1:1. We have decided to publish them.Therefore they can not be included graphs in the manuscript.

Unfortunately we don’t have the possibility to performe EDAX characterization in the short time given. Anayway it must be considered that EDAX response very well for transition metals because high count rates are allowed, and the system can detect extremely low concentrations of them.  Otherwise for light elements such as carbon, nitrogen and oxygen atoms much lower count rates are generated, making it difficult to detect these atoms at low concentrations. Usually, in the  best cases EDAX detects an element present in a sample at concentrations  between 1000 ppm and 5000 ppm, comparabile with high resolution XPS and AES  detection limits. In particular XPS has a sensitivity threshold detection limit of 1 at% and in this work it was employed to assess the purity of the sample investigated

(3) The format errors were eliminated and the figures’presentation notevolly improved

(4) Peak fitting is the procedure to separate the undistishble components under an experimetal peak shape. The fit function is the function empolyed to generate the synthetic components not directly visible on the experimental spectrum. Deconvolution is the presence of the overall synthetic components obtained by peak fitting procedure.

You are right for the misunderstandding, because withfit we include the three saparate meaning, according to the contex. We hope this misunderstanding was eliminated.

Best Regards

The Authors

Reviewer 2 Report

The reviewed paper concerns the application of the XPS method in determining the chemical composition and homogeneity of the produced ZrN layers. Nevertheless, the publication requires many additions and corrections before being published:

First, the abstract is too long and too detailed. The abstract should contain only key information for the article, such as: a brief description of the purpose of the work, the material tested, the measurement technique used (without details), a short summary of the most important results. The current abstract does not cover the content of the work.

Secondly, there are many typos in the text. The publication should be carefully corrected in linguistic terms.

Thirdly, the purpose of the work is not very precise, whether it generally refers to the determination of the chemical composition of ZrN layers using the XPS method or the use of appropriate procedures for deconvolution of the peaks leading to the determination of layer homogeneity.

Fourthly, there is no information about what is a new in the work in relation to the XPS method, which is normally used in this type of measurement.

Fifth, did the Authors use any charge compensation system during the XPS measurements?

Sixth, usually high-resolution XPS spectra are shown in publications in the range of binding energies from highest to lowest value, and not vice versa [Surface Science Spectra 5, 152 (1998); https://doi.org/10.1116/1.1247861].

Seventh, in general, the units on the X-axis (BE) for high resolution XPS spectra (Zr3d, N1s) should be scaled by 2 electrons to better visualize the described changes.

Eighth, Figure 2, the Authors assigned the peak at 179.59 eV to ZrN, which is in line with expectations [Surface Science Spectra 5, 152 (1998); https://doi.org/10.1116/1.1247861, Surface & Coatings Technology 202 (2008) 3129–3135 ], and the peak at about 182.0 eV to the ZrN satellite, which is an unusual interpretation. According to the literature information, the appearance of the so-called line muliplet splitting or shake-up for Zr (182 eV) is surprising [Moulder, J. F.; Stickle, W. F.; Sobol, P. E.; Bomben, K. D. Handbook of X-ray Photoelectron Spectroscopy, a Reference Book of Standard Spectra for Identification and Interpretation of XPS Data; Chastain, J., King, R. C., Jr., Eds.; Physical Electronics Inc.: Eden Praire, Minnesota, 1995] .The spectral shape of Zr3d is more suggestive of another chemical state, ZrO2 oxide [Nature Materials, VOL 19, March 2020, 282–286; Journal of the European Ceramic Society 18 (1998) 1293-1299, Thin Solid Films 520 (2012) 3532–3538]. The Authors themselves mentioned that they omitted the influence of oxygen when interpreting. Therefore, what is the physical backround for such an interpretation. This remark applies to all Zr3d spectra shown in the publication.

Ninth, what is the influence of oxygen on the formation of ZrN layers, in particular in near-surface zones. Zr has a relatively high affinity to oxygen.

Tenth, the Authors of the work should present high-resolution spectra for Zr3d, N1s and O1s before the ion etching process and after specific cycles, so that the spectra can be compared and see whether the effect of ion mixing occurs despite the use of a cluster gun. Does this effect influence on the position of the satellite peak and the formation of non-stoichiometric zirconium nitrides?

Eleventh, the Authors of the paper emphasized in the paper that it is difficult to distinguish the different forms of zirconium nitrides based on the Zr3d spectrum, with which the reviewer agrees. Therefore, how the ratio of Zr to N changes in the depth of the examined layers. This would allow to determine the homogeneity of the produced layers.

Author Response

Answers to Referee 2 Ref. Manuscript ID applsci-2041668

Kind Referee 2,

thank you for your contribution. We acceped all the corrections/suggestions you listed in the format report.

We operated major revisions in the manuscript, also to consider the comments of other refeers

In what follows the authors are going to reply point by point to you.

  • (1) The title was changed and shortened
  • (2) Abstract was shortened at 388 words, always a bit long respect lenghs of 150-200 words, but the substantial changes operated turns out fruitful to readers’ comprehension. The abstract now reflects better the key points developed in the paper.
  • (3)Format of the text and abnormal spaces were eliminated
  • (4) The Introduction was rivisited improving both the English style but also the content regarding the importance and particularity of quantitative XPS for ZrN films
  • (5)Tables were reformatted. The parentheses would have been indicated the three groups of samples investigated. We used another graphycal method to separate them in the table. The parenthses were delated.
  • (6) This is the most important obsevation of referee 2 which allowed to introduce a new paragraph (3 page 6) entitled “Preliminary preparation procedures for XPS experimental spectra acquisitions and definition of quantitative parameters of the research”. We hope Referee 2 finds this paragraph a big improvement for the quality of the whole paper, that is the authors’ opinion.
  • (7)Misspelling errors and English style have been checked
  • (8) and (9) Except statistical analyses that we are considering for the next publication we tryed to define very well the parameters for a quantitative and accurate study by XPS of ZrN films, as reported in the title. Every step that drives to quantification of data was explaned this time. The errors in area calculatioons were minimized and this was underlined in Table IV (newly introducted at page 14) where a comparison of the results obtained with two different fit-functions of the sofware SLG(30) and LA(a,b,c) has been reported and commented.To ensure the final  results were actually accurate we took as reference parameter the output value after peak fit optimization that is The Residal standard deviation. The lower it is the best is the matching between the fie evelope and the experimentalpeak shape.
  • (10) Figure 9 was eliminated because not easy to change and expecially unseful because it was object of a marginal theme that i show the tail trend changes increasin the level n of sputter erosion.
  • (11) The data analysis was completely revised, offering now not only clearness for the readers in the field but also a logical presentation for every kind of reader.

Best Regards

The Authors

Reviewer 3 Report

1)         Title is too long to highlight the research spots

2)         Abstract is too long and is poor in logic, such as the 1st sentence being too long to bring in puzzling for readers. Abstract length is usually 150~200 words. Reduction of enough words is necessary for the focused points. Of course, I have no additional words for paper writing skills because I am on the way too. Nevertheless, the customary regulations may be useful for most researchers and postgraduate students.

3)         Many abnormal spaces exist in the current manuscript, such as P1 Line40, P2 Line55, P4 Line115, Line135, etc.

4)         Introduction section is not enough and needs to improve for presenting the importance and particularity of quantitative X-ray photoelectron spectroscopy for ZrN thin films.

5)         Tables in the research paper should be continuous and are not permitted on two separate pages. Why do you use the big parentheses?

6)         Before and duration of XPS characterization, how to prepare and conduct these experiments for the X-ray Photoelectron Spectroscopy, and how to make sure that enough quantitative precision.

7)         There are enough misspelling errors in the current manuscript, please check it carefully.

8)         As a quantitative research paper, how to realize the definition of quantitative parameters, and is there any normalization process or quantitative reference for parameters? The definition of the characteristic parameters of the research object is necessary, and the statistical analysis should also be necessary.

9)         How to ensure the results whether or not from the measured data rather than systemic errors, such as the small differences for narrow regions in Table III. The differences among the data sets are unbelievable, especially for the evolved mathematical subtraction and other data processing.

10)       Mathematical subtraction process in Fig.9 is not clear to readers, please provide enough information on the adopted theoretical method(s).

11)       The data analysis in the results section is not enough and clear, which is harmful to the readers’ interest.

Author Response

Answers to Referee  3 Ref. Manuscript ID applsci-2041668

Kind Reviwer 3,

Fist of all the authors want to say they  acceped all the comments/corrections suggested  and cosequently improved/corrected the original manuscript, introducing also new elements and data to answer to some points (expecially  eighth and tenth) .Anyway we are going to comment point by point the Review Text Format, we received from you. Finally English language, style and minor spell checks were improved.

  • First Point: The abstract was shorted (388 words now) and modified in order toc over the content of the work;

  • Second Point English mistakes and linguistic terms were corrected. Sometimes the whole sentence was re-written and many ripetions were eliminated;

  • Third Point The purpose of the work was widely explained and the English level and style improved, in order to be clearer in the intentions and what has been done.

The use of an appropriate procedure for paeks’ deconcovolutions let us to define quantitatively some characteristics (microanalytical) of the samples.Please refer to Paragraph 3 where is (hopefully) reported the Preliminary preparation procedures for XPS (spectra acquisitions) and it is defined the quantitative parameters of the research;

  • Fourth poin. The novelty of the paper is clearly stressed and does not relates to XPS method by itself, in fact it is widely employed all over the world, but to the good quality of deposited samples ( loking forword many working years) -once for all the complete absence of oxygen in the films-which allowed to realize quantitative XPS elaboration of raw data ( including satellites)  that at our best knowledge is an original result and useful from the applications point of view (please read Discussions at pages 16-17).

  • Fifth point . During the XPS measurements 9/10 samples presented no need of charge compensation. You can refer in the text to Figures 9a) and b) where at page 12 paragraph 3.7 it is written  “[…]the raw data without any charge correctons or manipulation”;

  • Sixth Point. We agree, in fact in the PHI handbook the energy range on X-axis goes from high to low values. Anyway it is a convention and we decided to left the X-axis BE in our graphs from low to high energy values as in previous papers we published. We hope this is not a problem for you.

  • Seventh Point. If we are with you for this comment, if there is no charge compensation to make manually it is not right to scale of 2 eV the values read on the X-axis “to better visualize high resoltion spectra”. For this reason we do not applied this suggestion.

  • Eight Point Doubs arose in attributing the BE position the Zr 3d5/2 satellite peak at around 182 eV are eaili unravaled. Our XPS measurements start from the absence of oxygen in the film and therefore the peak position read after fit is not influnced by other componds as in the dated references you referred to. We can believe that the satellite peak position could seem “ an unsual interpretation” but we are take confort from a recently published paper [Applied srface science 396 (2017) 347-358 G. Greczynki et al “Corelevel spectra and binding energies….” Applied Surface Science ] where referring to figure 6b at page 354 of the paper it is reported (they say for the first time)Zr 3d- satellites  presented after deconvolution and in particular the 3d 5/2 (correspodent line) falls at 182.7 eV even a bit higher than ours.

  • Ninth Point. Zr posses a high affinity to oxygen, but in the samples investigated apart from sample ZrN A which is not a good result for deposition because there is a very little percentage of oxigen contamination in-depth, all the 9 remaining samples are oxygen free. Therefore inthese samples oxygen forms only a very thin cap-layer of oxides, but inside it is not present. This finding is not a case. We had a plasma spectrometer in the deposition chamber too monitor the undesired chemical species and also some other optionals (cryopum and warming of the targets) all installed in order to eliminate the oxygen, and adventitios contaminations as well (i.e acrbon).

  • Tenth Point The correction was completely shared and therefore we inserted the two new figures 9 (a) and 9 (b).It allowed to define the influnce of sputtering in the high resolution spectra. Please also consider the quantitative table IV at page 14 (newly iinsered) to better compare the N/Zr ratio at different sputtering levels. The sputtering at our working conditons does not influence the stoichiometry.

  • Eleventh Point The homogeneity was already touched in the previous point, indicating the new Table IV. In this table it is also shown that the fit function LA(a,b,c) gives output fits with a very low margin of errors respect to the fit function SLG(30) we used to consider in orevious reserch works.

Best Regards

The Authors

Round 2

Reviewer 2 Report

The reviewed paper has been carefully revised and many comments have been included in the revised version. Nevertheless, there will still be doubts about the interpretation of the Zr3d spectra, in particular the defined satellite lines. Such an interpretation is unheard of in the literature. It results from the proposed deconvolution model of XPS spectra, which proved to be excellent in determining the stoichiometry of the produced compounds. In this case, the Authors of the work should add a paragraph to the discussion of the results on the basis of the formation of satellite peaks, in particular based on literature data, an excellent example being metal oxides. It's known that „Satellites arise when a core electron is removed by a photoionization. There is a sudden change in the effective charge due to the loss of shielding electrons. (This perturbation induces a transition in which an electron from a bonding orbital can be transferred to an anti-bonding orbital simultaneously with core ionization). Therfore, two types of satellite are detected: shake-up or shake-off”. Then relate these literature data to the experimental data for ZrN. It would be good to show the wide scans and oxygen spectra recorded on the surface of the produced coatings without ion etching (state 0) as supplementary materials. The surface in as-received state will be crucial in determining the chemical state of the elements detected in the samples.

Author Response

Answers to referee 2 MAJOR REVISIONS

Kind Reviwer 2,

first of all, thanks for the shareable comments about satellites peaks studied by XPS in metal oxides (ex stable oxides of Ni 2p Ce 2p Fe 2p, Cr 3d  etc)..

  • We decided to introduce a new paragraph (3.8) in RESULTS section. A different situation must should be considered for metal transition nitrides: where a different situation emerges, respect to insulators, according to their electronic structure deduced by first-principles studies. Testually it has been stated: “First-principles simulations were used to determine electronic structures, vibrational spectra, and thermal properties of ScN, ZrN, and HfN in the rock salt […] Calculated band structures of ScN, ZrN and HfN along high symmetry directions of the Brillouin zone,indicate that ZrN and HfN have similar band structures and both are metallic in nature.[…]

We improvede the references both in number and for subject ( deposition techniques of ZrN for the introduction and 5more for th e3.8 subparagraph abot satellites. Consequently we had to improve Discussions and Conclusion paragraphs.

This preface is useful to say that in our c_ZrN films, metallic in nature, there is the possibility to observe satelites peaks.

Second point t related to the comparison of surface spectra (figure 9 a in the text) and after sputtering (level 2) Figure 9b. We changed the manuscript, improving the quaity under every point of view ( English language, clearness etc)

 Regarding the third point “moving the surface spectra of figure 9 a togheter with their survey scans in a supplementary file”it  is surely a good idea also to shorten the paper. Nonetheless it turns out important to leave  figure 9 a together with figure 9b without surveys scans, to leave the reader making a quick comparison between the two situations.

Best Regards

The Authors

Mirenghi L. and Rizzo A.

Answers to referee 2 MAJOR REVISIONS

Kind Reviwer 2,

first of all, thanks for the shareable comments about satellites peaks studied by XPS in metal oxides (ex stable oxides of Ni 2p Ce 2p Fe 2p, Cr 3d  etc)..

  • We decided to introduce a new paragraph (3.8) in RESULTS section. A different situation must should be considered for metal transition nitrides: where a different situation emerges, respect to insulators, according to their electronic structure deduced by first-principles studies. Testually it has been stated: “First-principles simulations were used to determine electronic structures, vibrational spectra, and thermal properties of ScN, ZrN, and HfN in the rock salt […] Calculated band structures of ScN, ZrN and HfN along high symmetry directions of the Brillouin zone,indicate that ZrN and HfN have similar band structures and both are metallic in nature.[…]

We improvede the references both in number and for subject ( deposition techniques of ZrN for the introduction and 5more for th e3.8 subparagraph abot satellites. Consequently we had to improve Discussions and Conclusion paragraphs.

This preface is useful to say that in our c_ZrN films, metallic in nature, there is the possibility to observe satelites peaks.

Second point t related to the comparison of surface spectra (figure 9 a in the text) and after sputtering (level 2) Figure 9b. We changed the manuscript, improving the quaity under every point of view ( English language, clearness etc)

 Regarding the third point “moving the surface spectra of figure 9 a togheter with their survey scans in a supplementary file”it  is surely a good idea also to shorten the paper. Nonetheless it turns out important to leave  figure 9 a together with figure 9b without surveys scans, to leave the reader making a quick comparison between the two situations.

Best Regards

The Authors

Mirenghi L. and Rizzo A.

Answers to referee 2 MAJOR REVISIONS

Kind Reviwer 2,

first of all, thanks for the shareable comments about satellites peaks studied by XPS in metal oxides (ex stable oxides of Ni 2p Ce 2p Fe 2p, Cr 3d  etc)..

  • We decided to introduce a new paragraph (3.8) in RESULTS section. A different situation must should be considered for metal transition nitrides: where a different situation emerges, respect to insulators, according to their electronic structure deduced by first-principles studies. Testually it has been stated: “First-principles simulations were used to determine electronic structures, vibrational spectra, and thermal properties of ScN, ZrN, and HfN in the rock salt […] Calculated band structures of ScN, ZrN and HfN along high symmetry directions of the Brillouin zone,indicate that ZrN and HfN have similar band structures and both are metallic in nature.[…]

We improvede the references both in number and for subject ( deposition techniques of ZrN for the introduction and 5more for th e3.8 subparagraph abot satellites. Consequently we had to improve Discussions and Conclusion paragraphs.

This preface is useful to say that in our c_ZrN films, metallic in nature, there is the possibility to observe satelites peaks.

Second point t related to the comparison of surface spectra (figure 9 a in the text) and after sputtering (level 2) Figure 9b. We changed the manuscript, improving the quaity under every point of view ( English language, clearness etc)

 Regarding the third point “moving the surface spectra of figure 9 a togheter with their survey scans in a supplementary file”it  is surely a good idea also to shorten the paper. Nonetheless it turns out important to leave  figure 9 a together with figure 9b without surveys scans, to leave the reader making a quick comparison between the two situations.

Best Regards

The Authors

Mirenghi L. and Rizzo A.

Please find enclosed the answers to your corrections/observations. The manuscript was changenged, consequently

Thank you for the collaboration.

Best Regards

The Authors

,

,

,

Reviewer 3 Report

The improvement of the paper is obvious, though some issues are not the expected result. Thanks for your revision.

(a) A whole praphgy for the Introduction would be prone to reduce the logic of the whole manuscript.

(b) Please make sure that all (sub)titles are right, especially for the presentation of the RESULTS. 

(c) Enough figures are not uniform and are not aligned well, such as Big blacks filled with Small charts.

(d) Figure 9 is not enough clear for a regular research paper.

Author Response

Please find enclosed the answers to your corrections/observations. The manuscript was changenged, consequently

Thank you for the collaboration.

Best Regards

The Authors
